# Impact of Endoscopic Ultrasound-Guided Tissue Acquisition on Decision-Making in Precision Medicine for Pancreatic Cancer: Beyond Diagnosis

**DOI:** 10.3390/diagnostics11071195

**Published:** 2021-06-30

**Authors:** Hiroshi Imaoka, Mitsuhito Sasaki, Yusuke Hashimoto, Kazuo Watanabe, Shoichi Miyazawa, Taro Shibuki, Shuichi Mitsunaga, Masafumi Ikeda

**Affiliations:** Department of Hepatobiliary and Pancreatic Oncology, National Cancer Center Hospital East, 6-5-1, Kashiwanoha, Kashiwa 277-8577, Chiba, Japan; mitsasak@east.ncc.go.jp (M.S.); yushashi@east.ncc.go.jp (Y.H.); kazuowat@east.ncc.go.jp (K.W.); smiyazaw@east.ncc.go.jp (S.M.); tshibuki@east.ncc.go.jp (T.S.); smitsuna@east.ncc.go.jp (S.M.); masikeda@east.ncc.go.jp (M.I.)

**Keywords:** endoscopic ultrasound-guided tissue acquisition, endoscopic ultrasound-guided fine needle aspiration, endoscopic ultrasound-guided fine needle biopsy, next-generation sequencing, precision medicine, pancreatic cancer, chemotherapy, biomarker, targeted genome sequencing

## Abstract

Precision medicine in cancer treatment refers to targeted therapy based on the evaluation of biomarkers. Although precision medicine for pancreatic cancer (PC) remains challenging, novel biomarker-based therapies, such as pembrolizumab, olaparib, and entrectinib, have been emerging. Most commonly, endoscopic ultrasound-guided tissue acquisition (EUS-TA) had been used for the diagnosis of PC until now. However, advances in EUS-TA devices and biomarker testing, especially next-generation sequencing, have opened up the possibility of sequencing of various genes even in limited amounts of tissue samples obtained by EUS-TA, and identifying potential genetic alterations as therapeutic targets. Precision medicine benefits only a small population of patients with PC, but biomarker-based therapy has shown promising results in patients who once had no treatment options. Now, the role of EUS-TA has extended beyond diagnosis into decision-making regarding the treatment of PC. In this review, we mainly discuss tissue sampling by EUS-TA for biomarker testing and the current status of precision medicine for PC.

## 1. Introduction

Pancreatic cancer (PC) is one of the deadliest cancers known and the fourth leading cause of cancer death. It is estimated that, in 2017, approximately 441,000 patients died of this disease around the world, which corresponds to an increase in the mortality from this cancer by 2.3-fold since 2003 [1]. The disease carries a dismal prognosis, with a 5-year survival rate of 10%, which is the lowest known for any cancer types [2]. Furthermore, there has also been little improvement in the prognosis of PC over the years. One epidemiological report based on the Netherlands Cancer Registry data concluded that the overall survival (OS) of patients with PC had improved by only 0.7 months in the past 20 years [3].

There are two major possible reasons for the poor prognosis of PC. One is the difficulty in early diagnosis of this cancer. The absence of specific symptoms usually results in late diagnosis, with the disease often already having advanced by the time of diagnosis. Furthermore, both serologic evaluation and abdominal imaging have disadvantages in identifying tumors in an early stage. Probably, carbohydrate antigen 19-9 (CA19-9) testing and abdominal computed tomography (CT) are the most widely used tools included in the primary diagnostic approach for PC in suspected cases. The reported sensitivity and specificity of CA19-9 for the diagnosis of PC range from 70–90% and 90%, respectively [4,5,6,7,8,9]. However, measurement of the serum CA19-9 level is of limited sensitivity for the diagnosis of small-sized cancers. In one large screening study that included 70,940 asymptomatic individuals, the positive predictive value was very low (only 0.9%) [10]. In a meta-analysis conducted to compare the diagnostic efficacy of multiple imaging tests for the diagnosis of PC, CT showed a high pooled sensitivity and specificity of about 90% [11]. However, the accurate diagnosis of small PC by CT is still a challenge, with a reported sensitivity of 58–77% for the detection of small (≤20 mm) tumors [12]. Endoscopic ultrasound (EUS), which provides high-resolution images, has a complementary role with CT for identifying small tumors and assessing the presence/absence of vascular invasion [13,14]. Furthermore, EUS guidance also enables the safe and easy acquisition of tissue samples; thus, “endoscopic ultrasound-guided tissue acquisition” (EUS-TA) has now become a standard approach for the diagnosis of PC.

The other reason is chemoresistance of PC. Molecular-targeted therapies and immune checkpoint inhibitors (ICIs), whose efficacies have been highlighted for various kinds of cancers [15,16,17,18], are of limited value in PC [19,20,21,22]. PC is characterized histopathologically by a fibrosis-rich stroma and macrophage infiltration. This so-called “tumor-microenvironment” not only prevents the infiltration of chemotherapeutic agents into the tumor but also protects the tumor tissue from immune response and promotes tumor growth by expressing growth factors [23,24,25]. Thus, PCs are called immunologically “cold tumors” that have been shown to be refractory to ICI therapy.

Precision medicine refers to the tailoring of treatment based on an individual’s genetics, lifestyle, and environment [26,27]. In cancer treatment, precision medicine refers to targeted therapy based on biomarkers, including proteins and genes: for example, *EGFR* in lung cancer [28] and *BRCA1/2* in ovarian cancer [16]. Although precision medicine has drawn much attention recently, precision medicine for PC still remains a challenge. In the carcinogenesis of PC, *KRAS* mutations are first triggered, followed by mutations in tumor suppressor genes such as *CDKN2A*, *TP53*, and *SMAD4*. Mutations of these four genes are seen at a high frequency in PC, and these are referred to as driver gene [29]. However, none of these driver gene inhibitors have reached the stage of clinical application, despite intensive efforts.

At present, the importance of next-generation sequencing (NGS) for precision medicine is gaining attention. NGS enables the sequencing of various genes even in the limited amounts of samples obtained by EUS-TA and allows potential genetic alterations as therapeutic targets to be identified. Thus, obtaining adequate tissue samples is mandatory for NGS, and the role of EUS-TA has now extended beyond diagnosis to decision-making in the treatment of PC. Thus, not only oncologists, but also endoscopists, need to arm themselves with knowledge about the current status of precision medicine for PC and the role of EUS-TA in precision medicine. In this review, we summarize the recent advances in the application of EUS-TA to biomarker testing and the current status of precision medicine for PC based on these biomarkers.

## 2. Endoscopic Ultrasound-Guided Tissue Acquisition (EUS-TA)

Previously, the diagnostic performance of EUS-TA was mainly evaluated based on the sensitivity and specificity of cytological diagnosis of malignancy. For the diagnosis of a solid pancreatic mass, EUS-TA has been proven by meta-analyses to be highly accurate (pooled sensitivity 85%–89% and specificity 96%–100%) [30,31,32]. A recent prospective study showed that the sensitivity of endoscopic ultrasound-guided fine needle aspiration (EUS-FNA) for the diagnosis of a solid pancreatic mass is over 90% [33,34]. In addition to such conventional pathological assessment, the importance of ancillary studies, such as immunohistochemistry (IHC) and molecular analyses, has also been emphasized for a long time.

Many studies have already indicated that the addition of molecular/genetic analyses, such as the identification of the *KRAS* mutations, to cytological examination may improve the diagnostic sensitivity for PC, especially in cases in which the pathological diagnosis proves inconclusive [35,36,37]. In a meta-analysis conducted by Gillis et al., the pooled sensitivities of cytology and *KRAS* mutations for the diagnosis of a pancreatic mass were 0.806 and 0.768, respectively, but when the two tools were used in combination, the diagnostic sensitivity increased to 0.887 [38]. However, in the past, the roles of both tools were limited to being complementary to histopathological diagnosis. Now, with the emergence of precision medicine, the significance of *KRAS* mutations is changing, as *KRAS* itself is coming to be recognized as a therapeutic target in PCs. For example, sotorasib showed promising antitumor activity in patients with advanced solid tumor harboring *KRAS^G12C^* in a phase 2 trial [39]. Thus, the therapeutic strategy for PC is changing dramatically as novel chemotherapies based on biomarkers, such as pembrolizumab [19], have come to be approved.

Newer EUS-TA needles may enable the acquisition of adequate core tissue samples for biomarker testing [40], but the total amount of sample obtained by EUS-TA is limited. Although advances in molecular biology have enabled the detection of biomarkers even in limited amounts of samples, the acquisition of relatively large tissue samples, while avoiding contamination, is mandatory for genomic analyses. The effects of various factors on the quality of tissue samples obtained by EUS-TA have been identified. These factors are described below in greater detail.

### 2.1. Needle Type (EUS-FNA vs. EUS-FNB)

Previously, EUS-FNA needles were developed with a focus on how to collect as many specimens as possible. The EUS-FNA needle, a needle with a simple structure used in combination with suction, is the most widely used needle for the diagnosis of PC, with a reported high sensitivity for the diagnosis of approximately 90% [30,31,32,41]. However, diagnostic failures may be caused by various factors, including insufficient sample volume, loss of histological structures, and the need for IHC. Therefore, endoscopic ultrasound-guided fine needle biopsy (EUS-FNB) needles began to be developed to preserve the tissue architecture and improve sample adequacy and diagnostic accuracy. The first developed EUS-FNB needle was stiff and difficult to use. However, newer generation EUS-FNB needles were developed and improved on sample adequacy and diagnostic accuracy [42,43,44]. In brief, EUS-FNA needles have a simple structure with an inner lumen and allow the collection of cells by aspiration with a syringe. On the other hand, EUS-FNB needles have a complex tip structure, which allows core samples to be collected by shearing the tissue from the target lesion [40] (Figure 1). Although the diagnostic benefit of EUS-FNB needles had been debated for a long time [41,44,45,46,47,48,49,50,51,52,53], the diagnostic accuracy of third-generation EUS-FNB needles has shown superiority in accuracy in recent randomized controlled trials [54,55]. Furthermore, the FNB needle showed a higher histologic core procurement rate than the FNA needle [54]. This is also true for pancreatic neuroendocrine tumors [56,57]. To date, the superiority of the EUS-FNB needle in tissue sample quality has been widely accepted [58]. Furthermore, considering tissue sample quality, the EUS-FNB needle may be more suitable for biomarker testing, as shown in the randomized trial [59] and retrospective studies [60,61].

### 2.2. Needle Size

Obviously, cellularity in the specimen is important for biomarker testing. Furthermore, the tumor fraction is also important for successful NGS. Samples with low cellularity are associated with an increased risk of insufficient DNA and polymerase chain reaction (PCR) failure, whereas samples with low tumor fractions are associated with an increased risk of false-negative results [62]. Theoretically, larger needles allow the collection of larger samples. Commercially available EUS-TA needles range in size from 19–25 gauge. The 25-gauge needle, the thinnest needle of them all, is flexible and easy to use. Although the amount of sample obtained is relatively low with this one, it has the advantage of being less contaminated [63]. On the other hand, a 19/20-gauge larger needle is stiffer and more difficult to handle, and its use is potentially associated with a high risk of complications. Furthermore, larger needles may cause greater contamination with blood, necrosis, and fibrous stroma, resulting in a lower tumor fraction. Based on these findings, the intermediate 22-gauge needle is considered the most versatile and appropriate for obtaining tissue samples for genomic analyses. Park JK et al. reported from their retrospective study how larger needles allow the retrieval of a higher quantity of DNA obtained by EUS-TA [64]. They performed sampling using a FNA needle or a reverse bevel needle (19, 22, or 25-gauge), and DNA was extracted from the frozen tissue sample. The NGS success rate was 57.4%. The DNA quantity obtained with a 25-gauge EUS-TA needle was significantly smaller than that obtained with a 19- or 22-gauge needle (*p* < 0.001). A multivariate logistic regression model only identified needle size as a significant factor influencing the success rate of NGS (a 19- or 22-gauge needle versus a 25-gauge needle: odds ratio, 2.29; 95% CI, 1.08 to 4.47).

### 2.3. Technical Aspects (Suction vs. Non-Suction Technique)

In contrast to the conventional suction technique using a syringe, the slow-pull technique, which was recently introduced for EUS-TA, minimizes the negative pressure, because the stylet itself is removed from the needle slowly and continuously. It was expected that the slow-pull technique would minimize blood contamination and allow higher-quality specimens to be obtained. However, the improved diagnostic performance of this non-suction technique still remains under debate. As compared to the conventional suction technique, there are reports that the non-suction technique is associated with a significantly lower rate of blood contamination [65,66] and higher cellularity [67]. On the other hand, many reports suggest the absence of any significant difference in the diagnostic performance between the two techniques [65,66,67,68,69,70]. One meta-analysis concluded that, although there was no difference in the accuracy or sensitivity between the two techniques, use of the non-suction technique was associated with a significantly lower rate of blood contamination compared to use of the conventional suction technique [71].

When performing EUS-TA, it is important to remain attentive to the need for obtaining tissue samples that would allow ancillary biomarker testing. In cases of metastasis, the use of a 22-gauge FNB needle and the non-suction technique may be optimal for tissue sampling. On the other hand, in EUS-TA for preoperative cases or cases of recurrence after surgery, the primary aim of EUS-TA is the diagnosis of malignancy, and biomarker testing is not necessary in tissue samples obtained by EUS-TA; surgical specimens should be used for biomarker testing, as they provide more adequate samples.

## 3. Biomarker Testing

Tissue samples obtained by EUS-TA are basically applied for the preparation of cytological smears and cell blocks. With cytological smears, diagnosis of malignancy is mainly made by Papanicolaou staining. Rapid on-site evaluation (ROSE) is also performed using Diff-quik staining to minimize the number of passes [72] and improve the diagnostic performance [73,74]. On the other hand, with cell blocks, histological assessment, and where necessary, biomarker testing is performed. The American Society of Clinical Oncology (ASCO) guidelines recommended early biomarker testing in PC patients who are likely to be potential candidates for precision medicine after first-line chemotherapy [75]. Widely used biomarker testing methods are IHC, fluorescence in situ hybridization (FISH), and NGS. While IHC and FISH are mainly used to identify single protein and gene alterations, novel genome sequencing technologies, such as NGS, are used to detect multiple genetic alterations simultaneously in relatively small amounts of tissue samples [76,77,78]. In IHC and FISH, the targeted protein or DNA in the sample is captured using a probe; then, these targets are visualized under the microscope using a dye or an enzyme bound to the probe. These methods allow analysis of a small number of tumor markers by searching for known “hotspots,” namely, genetic loci that are known to frequently show alterations. On the other hand, genome sequencing can determine the nucleotide order of a DNA fragment. Although traditional Sanger sequencing is widely used for DNA sequencing [79], it allows the sequencing of only one fragment at a time, making this method costly as well as time- and labor-intensive for large-scale sequencing. Furthermore, substantial amounts of DNA are required. NGS platforms allow the sequencing of a massively parallel collection of clonally amplified or single DNA molecules that are spatially separated in a flow cell [76]. This feature facilitates the sequencing of millions to billions of short fragments of DNA [80,81]. This is an important advantage that enables the screening of large numbers of samples with a short turnaround time. Furthermore, NGS technology requires a relatively small amount of DNA or RNA, in contrast to traditional sequencing technologies, along with a reduced overall cost of multiple-marker screening [77].

Several studies have examined the surrogacy of EUS-TA samples for surgically resected specimens in NGS. The results of these studies of NGS using EUS-TA samples showed high concordance with that in a surgically resected specimen used as a reference standard [82,83,84]. These findings are important because PC patients who potentially benefit from precision medicine are ineligible for surgical resection.

### 3.1. Immunohistochemistry

IHC, a familiar tool to clinicians, involves the use of an antibody as a probe. It specifically captures the targeted proteins and can also localize them in cancer cells. IHC used to examine protein expression has several advantages. It is commonly used in clinical laboratories and is therefore relatively straightforward to implement and validate. It also has the benefits of being inexpensive, requiring only a single unstained slide, and having a rapid turnaround time.

### 3.2. Fluorescence In Situ Hybridization

FISH involves the use of an oligonucleotide probe (consisting of relatively short base-pairs) labeled with fluorescent dyes to hybridize with the targeted DNA, and then be visualized with fluorescence microscopy. FISH can detect large structural variations at the DNA level and is often used in the clinical laboratory to detect oncogenic fusions in solid tumors. Advantages of FISH are that the amount of material required is only a few unstained slides—usually one unstained slide per probe examined—and that the turnaround time is usually only a few days [85].

### 3.3. Next-Generation Sequencing

NGS enables the sequencing of multiple genes in a limited number of samples obtained by EUS-TA. Recently, several studies have reported the adequacy of EUS-TA samples for NGS [61,64,84,86,87,88]. These reports are summarized in Table 1. Carrara S et al. reported, from their prospective study, that proper tissue specimens could be obtained by EUS-TA. They obtained samples with a 22-gauge Franseen FNB needle using the fanning and slow-pull technique and achieved successful DNA sample extraction and subsequent NGS in 97% of cases [88]. Young et al. reported on the performance of NGS from a retrospective review of sequencing performed in formalin-fixed paraffin-embedded (FFPE) samples obtained by EUS-TA. Using a customized gene panel (287 genes), genomic profiles were generated successfully from all 23 of 23 (100%) solid pancreatic masses, and the most common mutations observed were mutations of *KRAS* (78%), *TP53* (74%), *CDKN2A/B* (35%), *SMAD4* (17%), and *PTEN* (13%) [87].

Various factors can affect the results of NGS, including the process of library preparation, choice of assays, and the biomarker testing technique itself [89,90,91]. From the aspect of sample quality, two major factors are important to obtain successful results from NGS. One is the fixation/preservation of the tissue samples. FFPE specimens are still preferred for NGS, since FFPE specimens provide the advantage of direct evaluation of the tumor cellularity. Pathologists can select the appropriate area in the specimen for NGS, such as areas with high cellularity. Furthermore, FFPE samples allow the tissue architecture to be preserved and can be used for future ancillary studies. However, FFPE samples subjected to inappropriate fixation methods (i.e., inappropriate duration of fixation or concentration of fixative) may yield incorrect results [90,92]. Furthermore, DNA extracted from FFPE tissues is generally damaged, and long-term storage may also affect the DNA quality. Thus, it is preferable to perform NGS in FFPE samples that have been prepared less than three years ahead of the sequencing [89].

Another factor is the quality of the tissue samples. It is important to provide the optimal absolute number of neoplastic cells. Typically, tissue samples acquired by EUS-TA contain relatively fewer neoplastic cells as compared to surgical FFPE samples. In general, the frequency of successful NGS is lower in smaller specimens (EUS-TA) as compared with larger specimens (resection and excisions). Goswami RS et al. reported that the NGS success rate was lower in specimens obtained by FNA than in specimens obtained by resection (50% versus 97%, respectively; *p* < 0.0001) [93]. In their report, the extracted DNA concentration from larger tissue samples was significantly higher compared to that from smaller samples. However, an increase in the tissue sample size does not necessarily yield an increased NGS success rate. The tumor fraction, the percentage of tumor cells in the tissue sample, is more important than cellularity, when NGS is performed in small specimens. Increasing the tissue sample (nucleic acid) sometimes results in a decrease of the tumor fraction as a result of the inclusion of a greater number of non-tumor cells, such as stromal or inflammatory cells [91]. A low tumor fraction could potentially yield false-negative results; NGS commonly requires a minimum tumor fraction of about 10–20% [94,95].

The pathological sample quality is also important for the success of NGS. PC not only tends to show low cellularity but also to contain necrotic regions. Necrotic tissue contains cell debris, non-viable tumor cells, fragmented DNA, and few viable tumor cells [96]. These components may interfere with obtaining accurate results. Although anucleated blood cells, such as red blood cells, do not reduce the tumor fraction, the heme in the red blood cells can act as an inhibitor of the PCR reaction [97]. Therefore, special care is needed during the selection of tissue samples to enrich the tumor fraction, while minimizing the entrapped red blood cells in bloody specimens. Mucin and hyalinized stroma in tissue samples do not affect the PCR reaction itself; however, large quantities of these in tissue sections can reduce the overall tumor cellularity and dilute the samples [91].

FFPE specimens are frequently used for biomarker testing. Although the cellularity and tumor fraction are important for testing, the tumor fraction is generally low in PCs, since PC tissue contains more stromal cells, hematopoietic cells, or desmoplastic fibroblasts than tumor cells. In order to avoid a low-tumor fraction, the tissue samples obtained by EUS-TA should be divided among multiple blocks rather than being limited to a single block. On the other hand, in recurrent cases, surgical specimens are more suitable for biomarker testing because of their higher cellularity and tumor fraction. Therefore, biomarker testing should ideally be performed using surgical specimens.

There are 3 major types of NGS used for the identification of genomic alterations: whole-genome sequencing (WGS), whole-exome sequencing (WES), and targeted genome sequencing. Of these methods, WGS and WES are mainly used in the research field and are generally performed on blood or fresh frozen biopsy specimens. On the other hand, targeted genome sequencing is performed on both DNA (targeted DNA sequencing) and RNA (targeted RNA sequencing), with the former being mainly used in the clinical field. Tumor DNA can be extracted from FFPE specimens and then sequenced to investigate whether specific alterations are present in the tumor. One drawback of targeted genome sequencing is that, when novel structural variants are detected, it can be difficult to determine whether the event would result in a functional expressed fusion. Other drawbacks include the turnaround time, which is significantly longer than for IHC or FISH, and that a relatively large amount of material is required for the testing. On the other hand, a major advantage of targeted genome sequencing is that many genomic events can be interrogated, allowing for the simultaneous direct assessments of point mutations, insertions, deletions, copy number variations, and the tumor mutation burden, in addition to gene fusions.

#### 3.3.1. Whole-Genome Sequencing

WGS enables the most comprehensive study of entire genomes, which is valuable when seeking to discover novel genomic alterations. Since the entire genome is being sequenced, the noncoding sections of DNA within genes, called introns, can also be sequenced. Thus, performing WGS is a lengthy process that requires a large amount of sample input and is expensive. WGS typically requires 100–1000 ng of DNA, and it is generally not practical for testing the limited amounts of tissue samples obtained by EUS-TA. Thus, we recommend the use of more-focused targeted sequencing panels, to a limited number of specific genome alterations or RNA targets implicated in cancer pathways in clinical practice.

#### 3.3.2. Whole-Exome Sequencing

An alternative approach to WGS is to sequence only the exomes, called whole-exome sequencing, or WES. The exome consists of the exons, which are the coding parts of genes that are translated and expressed as protein; the exome comprises only about 2% of the whole genome. Exome sequencing allows researchers to focus on the part of the genome that is more likely to contain clinically actionable alterations, such as mutations of tumor-suppressor genes. However, it still requires a large amount of DNA (100–1000 ng). Compared to WGS, WES-generated data provide more confidence for detecting low-allele-frequency somatic alterations found in tumor tissue samples. However, it still produces large amounts of data that are irrelevant.

#### 3.3.3. Targeted Genome Sequencing

Targeted genome sequencing is the most widely used method for NGS. We can simultaneously analyze preselected target regions that include many known driver or clinically actionable genetic mutations. As compared to WGS and WES, targeted DNA sequencing is more flexible, since it allows the accurate and highly sensitive identification of the target genes to be balanced with the overall cost and data burden. Typically, targeted DNA sequencing requires a relatively small amount of DNA (10–300 ng). The great depth of sequencing coverage in targeted genome sequencing enables the profiling of even clinical samples with relatively low DNA quality and/or tumor content, such as FFPE specimens, and to detect genomic alterations that are present only in a small fraction of cancer cells. On the other hand, extra care needs to be taken during the analysis, especially for data generated from low-quality or fragmented DNA and/or without data from matched normal control tissue, because of the high depth of sequencing coverage. Ready-to-use NGS panels are commercially available for applications such as deep sequencing of single oncogenes (e.g., *KRAS*), as well as for emerging applications, such as monitoring the tumor mutation burden (TMB) and microsatellite instability (MSI).

Targeted RNA sequencing is also performed in the research field. Targeted RNA sequencing presents several advantages over targeted DNA sequencing. Detection of RNA-level fusions provides direct evidence that they are functionally transcribed, and analysis of the spliced sequence can determine whether the protein would be translated and in-frame. Fusion transcripts can also be detected with high confidence in the RNA of low-quality tumor samples, because gene fusions are often highly expressed in the tissue. One of the main drawbacks to working with RNA is its fragility. RNA can be extracted from FFPE tissue specimens, and targeted RNA sequencing generally requires 100–1000 ng of RNA. However, RNA is susceptible to fragmentation and degradation, especially in older tissue specimens, so extra attention needs to be paid to adequate quality control.

### 3.4. Future Perspectives

Future perspectives in NGS include the use of cytological specimens. Cytological smear specimens of EUS-FNA are rich in tumor cells, whereas FFPE specimens are often rich in tumor stroma. Recently, the excellent performance of cytological smear specimens in molecular analyses has been shown [98,99]. For lung cancer, the College of American Pathologists guidelines state that pathologists may use either FFPE samples (cell blocks) or other cytological preparations (cytological smears) as suitable specimens for biomarker molecular testing, including NGS [100]. In addition, FNA rinse sample and touch imprint cytology are promising approaches for NGS. EUS-TA samples are routinely rinsed and fixed in an alcohol-based fixative immediately after their acquisition, and this contributes to excellent preservation and quality of DNA and RNA. Wei et al. compared the performance of this FNA rinse samples and that of cell block samples used for NGS [101]. NGS was successfully performed using all samples, but much more DNA was obtained from the FNA rinse samples compared with the paired cell block samples (176.3 versus 10.6 ng/μL, respectively). Touch imprint cytology is a simple and cost-effective method: a tissue sample is touched onto a slide, leaving an imprint on the glass slide in the form of cells. Crinó et al. reported that the sample quality of touch imprint cytology was comparable to cytological smear specimens [102]. Touch imprint cytology allows obtaining both valuable cytological smears and histological samples in a single session of EUS-FNB. Thus, this method is a potentially useful method for NGS.

Another future perspective is to apply EUS-TA to the establishment of organoids. Organoids are three-dimensional self-assembled in vitro cultured structures derived from tissues and stem cells. They are widely used in various research fields, from basic developmental and stem cell research to personalized medicine. Recently, there have been increasing reports on the organoids’ establishment using tissue sample obtained by EUS-TA, and the success rate of organoids establishment ranged between 70 and 82% [103,104,105,106]. These patient-derived organoids have the potential to be used in drug screening and predicting the response to chemotherapeutic agents in pancreatic cancer patients in the future [107].

## 4. Precision Medicine for Pancreatic Cancer

Approximately 60% of PC patients have metastatic disease at diagnosis [3,108,109], and most patients are candidates for systemic chemotherapy. Chemotherapy for PC can be divided into two main categories: fluorouracil-based chemotherapy and gemcitabine-based chemotherapy. Among them, first-line treatment with FOLFIRINOX [110] (fluorouracil-based) and gemcitabine plus nab-paclitaxel [111] (gemcitabine-based) have been shown to increase the OS in patients with metastatic PC. However, the treatment needs to be discontinued in most patients, because of disease progression or the emergence of adverse events. Then, second-line chemotherapy is attempted, if the patient’s PS is good [112,113]. In the second-line setting, fluorouracil-based chemotherapy is recommended after previous gemcitabine-based chemotherapy, while gemcitabine-based chemotherapy is recommended after previous fluorouracil-based chemotherapy [113,114,115,116]. However, a phase III trial has clearly revealed only the efficacy of the nanoliposomal irinotecan/fluorouracil/leucovorin regimen in the second-line setting [117]. In general, median OS in first-line chemotherapy is about one year [110,111,118,119,120], and about six months in second-line chemotherapy [117,121,122,123] (Table 2). Precision medicine for PC is basically applied based on the evaluation of biomarkers after the failure of such standard chemotherapies. In this section, we provide an overview of precision medicine that has been approved in patients with PC.

### 4.1. Immune Checkpoint Inhibitors

ICIs have emerged as a new treatment paradigm for patients with many types of cancer [124,125]. Immune checkpoints are a normal component of the immune system that inhibit the immune response to protect the host from excessive immune responses, in order to maintain immune homeostasis. However, some cancer cells utilize this immune checkpoint mechanism to attenuate the immune response and survive [126]. ICIs bind to immune checkpoint proteins or their ligands and block these immunosuppressive signals. Although, until now, only a limited number of cancer patients have been shown to derive benefit from ICIs, MSI-high (MSI-H)/mismatch repair-deficiency (dMMR) is considered a biomarker for predicting the efficacy of ICIs [127,128] (Figure 2).

A microsatellites is a repetitive sequence of 1–6 nucleotides. As the number of repetitions increases, the gene or its product, the protein, tends to become unstable. If there is a base mismatch during DNA replication associated with cell division, the mismatch repair (MMR) mechanism works to repair it. MSI is the accumulation of microsatellite repeats in a genome because of the functional deficiency of MMR genes, such as in cancer cells, or because of defects in the process of replication repair. MSI-H is well-known to play an important role in the development and progression of cancers, such as Lynch syndrome [129,130]. On the other hand, MSI-H/dMMR tumors express more neoantigens newly derived from the genomic alterations in the cancer cells [131]. These neoantigens are easily recognized by the T-cells and induce an immune response [132].

Currently, approved ICIs targeting immune-checkpoint proteins are CTLA-4, PD-1, and PD-L1. Anti-PD-1 antibody, pembrolizumab, binds to PD-1 on the T cells and inhibits the interaction of PD-1 with the PD-L1 expressed in the cancer cells [133]. As a result, immunosuppressive signals to T cells are blocked, which allows T cells to attack cancer cells. In the KEYNOTE-158 trial, pembrolizumab showed efficacy in different tumor types harboring MSI-H/dMMR [19]. In the overall study population, the ORR and median PFS were 34.3% and 4.1 months, respectively. The frequency of MSI-H/dMMR in PCs is low, occurring approximately in only 1.0% [134,135]. However, 22 patients with PC harboring MSI-H/dMMR were included in this trial, and pembrolizumab showed some efficacy (ORR, 18.2%; median PFS, 2.1 months; OS, 4.0 months) in this population.

MSI analysis, IHC for MMR proteins, and NGS are used for MSI-H/dMMR testing. Both techniques showed high accuracy, over 90% [136]. MSI analysis is performed using DNA extracted from both the tumor and surrounding normal tissue in FFPE samples [137]. The usefulness of EUS-TA has been reported for MSI-H/dMMR testing. Sugimoto M et al. evaluated the adequacy of EUS-TA samples for MSI analysis in patients with PC [138]. Of a total of 89 samples (61 via EUS-FNA and 28 via EUS-FNB), EUS-FNB yielded a higher proportion of sufficient samples for MSI analysis compared to EUS-FNA (88.9% versus 35.7%, respectively; *p* = 0.03). NGS can also accurately assess MSI status in tumors [139,140], allowing for the comprehensive profiling of targeted genome sequencing, as well as MSI status by a single NGS.

Recently, TMB has been highlighted as another biomarker for predicting the efficacy of ICIs [141]. TMB is a measure of the number of somatic mutations per megabase (mut/Mb) in cancer cells. The TMB varies widely among tumor types, and PCs typically have a relatively low TMB (approximately 1 mut/Mb) [142]. Tumor with dMMR are characterized by a very high TMB (≥10 mut/Mb), and some studies have shown that, due to dMMR, cancer cells can produce heterologous antigens that are easily recognized by T cells [143]. TMB can be determined by using different methods; however, the optimal approach is the calculation of the mutational load based on WES encompassing approximately 30 Mb [144]. However, the commercially available, targeted DNA sequencing panel encompassing 0.8 Mb can also be used to estimate the TMB and has been validated [145]. Based on these findings, the US Food and Drug Administration (FDA) approved pembrolizumab for the treatment of refractory PC with MSI-H/dMMR or TMB-high.

### 4.2. PARP Inhibitors

Germline mutations of *BRCA1/2* are known to be associated with an increased risk of the development of ovarian cancer and breast cancer [146,147,148] and are also found in PCs at a frequency of about 4–7% [149,150,151,152,153]. Presence of germline *BRCA1/2* mutations is reported to be associated with enhanced antitumor effects of platinum drugs, such as oxaliplatin, which exert their antitumor effects by inducing the formation of DNA interstrand cross-links and DNA damage [154]. PARP inhibitors, such as olaparib, inhibit homologous recombination during the repair process after DNA damage and thus, exert their antitumor effects by inducing the apoptosis of tumor cells harboring germline *BRCA1/2* mutations. Olaparib has already been shown to be useful for maintenance treatment after platinum-based chemotherapy of ovarian cancer harboring germline/somatic *BRCA1/2* mutations [16,155].

In a phase III study (POLO trial), olaparib was demonstrated to show efficacy as maintenance therapy after first-line platinum-based chemotherapy in PC patients with germline *BRCA1/2* mutations [156]. The PFS, the primary endpoint of the trial, was 7.4 months in the olaparib group and 3.8 months in the placebo group (hazard ratio [HR], 0.53; 95% confidence interval (95% CI), 0.35–0.82).

On the other hand, the effects of PARP inhibitors against PCs harboring somatic *BRCA1/2* mutations are still unclear. However, one phase II study showed the efficacy of olaparib administered as monotherapy in previously treated PC patients with DNA damage repair gene (DDR) alterations other than germline *BRCA1/2* mutations [157]. The overall efficacy was modest (ORR, 2%; median PFS, 3.7 months; OS, 13.6 months). However, olaparib was demonstrated to exert efficacy in 24 patients with DDR phenotypes (14 with *ATM*, 3 with *ARID1A*, 2 with *PALB2*, 6 with other alterations). Another PARP inhibitor, rucaparib, has shown some activities in PC patients with a germline or somatic *BRCA1/2* mutations in phase 2 trials. Shroff et al. reported the efficacy of rucaparib in previously treated PC patients with a germline or somatic *BRCA1/2* mutations [158]. A total 19 patients were enrolled in this trial (16 with germline mutation, 3 with somatic mutation). Although subsequent enrollment was stopped due to futility, 2 out of the 3 patients harboring a somatic *BRCA2* mutation showed PR. Rucaparib was also demonstrated to show efficacy as maintenance therapy after first-line platinum-based chemotherapy in PC patients with germline or somatic *BRCA1/2* or germline *PALB2* mutations [159]. Rucaparib showed response in 3 out of 6 patients with a germline *PALB2* mutation, and 1 out of 2 patients with a somatic *BRCA1/2* mutation. One meta-analysis reported similar ORR of PARP inhibitors in patients with germline and somatic *BRCA1/2* mutations [160]. These findings indicate that PARP inhibitors may be effective in patients with a somatic *BRCA1/2* mutation and other DDR alterations, as well as in patients with a germline *BRCA1/2* mutation. In the near future, it may be meaningful to evaluate a somatic *BRCA1/2* mutation as a biomarker. In such a scenario, NGS using tissue samples would allow us to detect both germline and somatic *BRCA1/2* mutations. When *BRCA1/2* mutations are found by NGS in tumor-tissue specimens, it is necessary to distinguish between somatic and germline mutations [161]. For the identification of germline alterations, peripheral blood specimens are the most frequently used biological material. However, the variant allele frequency (VAF) reported by NGS can aid in distinguishing germline mutations from somatic mutations. A mutation is potentially a germline mutation if the VAF is approximately 50% (heterozygous) or 100% (homozygous). The VAF values associated with somatic mutations are usually lower, because these mutations are not present in all cells. However, somatic mutations may be associated with VAF values of over 50%, if the tumor fraction in the analyzed sample is high [162,163].

### 4.3. TRK Inhibitors

Members of the tropomyosin receptor kinase (TRK) receptor family, encoded by *NTRK1*, *NTRK2*, or *NTRK3*, are expressed in nervous tissue and are known to play roles in the development of the nervous systems and in pain perception [164]. Fusions involving these *NTRK* genes result in a TRK fusion protein that stimulates cell proliferation through the constant activation of signal transduction in cancer cells (Figure 3) [165]. These fusions are found at high frequencies in rare cancer types, such as secretory breast carcinoma [166] and mammary analogue secretory carcinoma of the salivary gland [167]. On the other hand, the frequency of *NTRK* fusion was low, less than 1.0%, in PCs [165,168].

Two TRK fusion inhibitors, larotrectinib and entrectinib, have been demonstrated to show promise. Drilon et al. reported the efficacy of larotrectinib against different tumor types harboring the *TRK* fusion gene [169]; a total of 55 patients were treated with arotrectinib: the ORR, the primary endpoint, was 75%, and the 1-year PFS rate was 55%. The subject population included 1 patient with PC, who showed partial response (PR).

Entrectinib is a potent and selective TRK and ROS1 inhibitor. In an integrated analysis of 3 phase I/II trials (the STARTRK-1, ALKA-372-001, and STARTRK-2 trials), entrectinib was revealed to be safe and effective against different tumor types harboring an NTRK fusion protein (ORR, 57%; median PFS, 11 months; median OS, 21 months) [170]. The subject population in these trials included 3 patients with PC. Of the 3 patients, 2 showed PR and 1 showed stable disease (SD) [171]. Thus, the FDA granted accelerated approval to larotrectinib and entrectinib for the treatment of PC harboring *NTRK* fusion. *NTRK* fusion is detected using NGS, FISH, or IHC [172].

### 4.4. Future Perspectives

Several promising targeted therapy based on biomarkers are currently being tested in clinical trials. Sotorasib irreversibly inhibits the function of KRAS G12C mutant protein by locking the protein in its GDP-bound inactive form [173]. In a phase II study (CodeBreaK100 trial), sotorasib was found to show promising efficacy in patients with different tumor types harboring *KRAS^G12C^* [39]. A total of 12 PC patients were enrolled in this trial, 1 patient showed PR, and 8 patients showed SD. Now sotorasib-based clinical trials, such as combination therapy with sotorasib and ICIs, are ongoing (ClinicalTrials.gov Identifier: NCT04185883).

Neuregulin-1 (NRG1) stimulates the proliferation, differentiation of mammary epithelial cells, glial cells, nerve cells, and muscle cells, and *NRG1* fusions result in the activation of various signaling pathways to promote cell proliferation [174]. The reported frequency of *NRG1* fusion in PC is as low as 0.13–0.4%; however, *NRG1* fusion is considered a driver gene in *KRAS* wild-type PC. Scharam, et al. reported their findings from an ongoing phase I/II basket trial of zenocutuzumab (MCLA-128) in patients with *NRG1* fusion-positive solid tumors [175]. Among 10 PC patients harboring *NRG1* fusion, ORR was 40% [176]. In the near future, precision medicine in PC will expand further.

## 5. Conclusions

Precision medicine still benefits only a small population of patients with PC. However, biomarker-based therapy has shown promising results in patients who once had no treatment options. With the advancement of precision medicine for PC, the impact of EUS-TA on therapeutic decision-making will become more significant. Thus, while performing EUS-TA, the operator should remain attentive to the need for obtaining tissue samples that would allow biomarker testing.

## Figures and Tables

**Figure 1 diagnostics-11-01195-f001:**
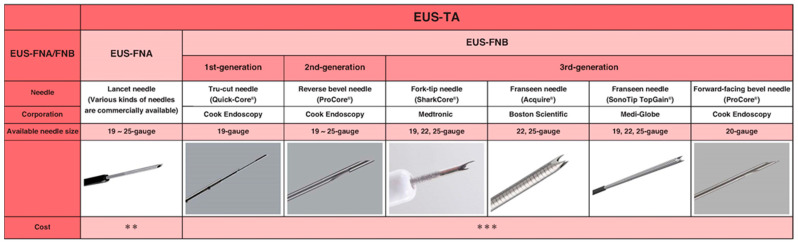
Overview of the needle design of EUS-FNA and -FNB needles.

**Figure 2 diagnostics-11-01195-f002:**
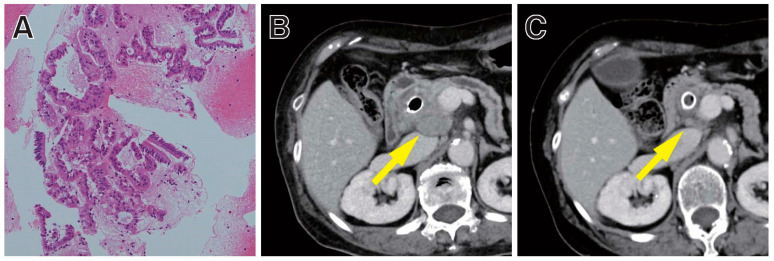
Clinical response to pembrolizumab in a PC patient with MSI-H. (**A**) Tissue sample from PC. MSI analysis using a FFPE sample obtained by EUS-FNB showed MSI-H; (**B**) The pancreatic head cancer was refractory to the previously administered systemic chemotherapy, with disease progression; (**C**) A durable and confirmed partial response was achieved. Substantial shrinkage in the size of the pancreatic head mass was noted even from early (by week 8, as shown; arrowed) in the course of therapy.

**Figure 3 diagnostics-11-01195-f003:**
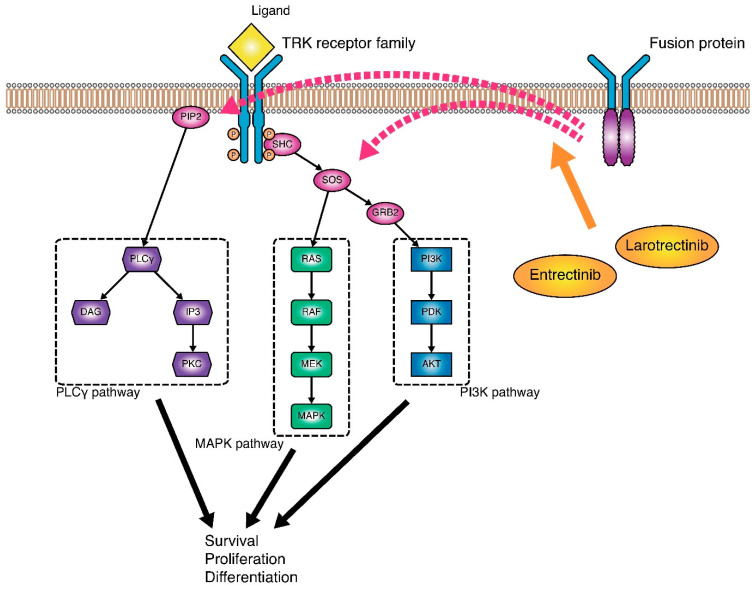
TRK pathway and mechanism of TRK inhibitors. The activation of TRK receptors (TRKA, TRKB, and TRKC) by the binding of the ligand neurotrophin promotes cell survival, proliferation, and differentiation through downstream PLCγ, MAPK, and PI3K pathways. Functional TRK-fusion proteins induce the ligand-independent activation of the tyrosine kinase domain to upregulate downstream cancer-related pathways. TRK inhibitors are tyrosine kinase inhibitors of TRK receptors and inhibit cancer-cell survival and proliferation by suppressing the MAPK pathway.

**Table 1 diagnostics-11-01195-t001:** Adequacy rate of EUS-TA samples for NGS (target genome sequencing) in pancreatic cancer.

Author (Year)	Study Type	No. of Patients	Type of Tumor	Needle Type (Needle Size)	Adequacy Rate for NGS	Required Tumor Fraction	Targeted Panel	Frequency of Genomic Alterations
Elhanafi S, et al. (2018) [61]	Retrospective study	167	PDAC	EUS-FNA/B	70.1%	≥10%	Custom panel (47 genes)	*KRAS* (88%), *TP53* (68%), *SMAD4* (16%)
		145		EUS-FNA (22-gauge)	66.9%			
		22		EUS-FNB (22-gauge)	90.9%			
Larson BK, et al. (2018) [86]	Retrospective study	61	Pancreatic exocrine malignancy	EUS-FNA/B	67.2%	≥20%	FoundationOne (315 genes)	NA
		7		EUS-FNA	42.9%			
		54		EUS-FNB	70.4%			
Young G, et al. (2013) [87]	Retrospective study	23	PDAC, Mucinous adenocarcinoma, adenocarcinoma NOS, PanNET	EUS-FNA	100%	≥20%	Custom panel (287 genes)	*KRAS* (78%), *TP53* (74%), *CDKN2A/B* (35%), *SMAD4* (17%), *PTEN* (13%)
Gleeson FC, et al. (2016) [84]	Retrospective study	47	PDAC, Ampullary adenocarcinoma, IPMN	EUS-FNA	61.7%	≥20%	Human Comprehensive Cancer GeneRead DNAseq Targeted Panel V2 (160 genes)	*KRAS* (93.1%), *TP53* (72.4%), *SMAD4* (31%), *GNAS* (10.3%)
Park JK, et al. (2020) [64]	Retrospective study	190	PDAC	EUS-FNA/B (19, 22, 25-gauge)	57.4%	≥30%	Custom panel (83 genes)	*KRAS* (93.1%), *TP53* (72.4%), *SMAD4* (31%), *GNAS* (10.3%)
Carrara S, et al. (2021) [88]	Prospective study	33	PDAC	EUS-FNB (22-gauge)	97%		AmpliSeq Comprehensive Panel v3 (161 genes)	*KRAS* (94%), *TP53* (78%), *SMAD4* (13%), *CDKN2A* (19%), *GNAS* (9%), *NOTCH2* (9%), *ATM* (9%)

NGS, next-generation sequencing; PDAC, pancreatic ductal adenocarcinoma; NOS, not otherwise specified; PanNET, pancreatic neuroendocrine tumor; EUS-FNA, endoscopic ultrasound-guided fine needle aspiration; EUS-FNB, endoscopic ultrasound-guided fine needle biopsy; NA, not available.

**Table 2 diagnostics-11-01195-t002:** Summary of pivotal trials of chemotherapy for pancreatic cancer.

Author (Year)	Trial	Extent of Disease	Treatment	n	ORR	Median PFS (Months)	Median OS (Months)	HR_OS_	*p*-Value
**First-line chemotherapy**									
Burris HA, et al. (1997) [118]		Locally advanced/metastatic	Gemcitabine versus 5-FU	63 vs. 63	0% vs. 5.65%	NA	5.65 vs. 4.41	NA	0.0025
Moore MJ, et al. (2007) [119]	NCIC CTC PA.3	Locally advanced/metastatic	Gemcitabine/erlotinib versus gemcitabine	285 vs. 284	8.2% vs. 6.9	3.75 vs. 3.55	6.24 vs. 5.91	0.82 (95% CI, 0.69–0.99)	0.038
Conroy T, et al. (2011) [110]	PRODIGE4/ACCORD11	Metastatic	FOLFIRINOX versus gemcitabine	171 vs. 171	31.6% vs. 9.4%	6.4 vs. 3.3	11.1 vs. 6.8	0.57 (95% CI, 0.45–0.73)	<0.001
Ueno H, et al. (2013) [120]	GEST	Locally advanced/metastatic	Gemcitabine/S-1 vs. S-1 versus gemcitabine	277 vs. 280 vs. 275	29.3% vs. 21.0% vs. 13.3%	5.7 vs. 3.8 vs. 4.1	10.1 vs. 9.7 vs. 8.8	0.96 (97.5% CI *, 0.78–0.1.18) 0.88 (97.5% CI *, 0.71–1.08)	<0.001 ^†^ 0.15 ^‡^
Von Hoff DD, et al. (2013) [111]	MPACT	Metastatic	Gemcitabine/nab-paclitaxel versus gemcitabine	431 vs. 430	23% vs. 7%	5.5 vs. 3.7	8.5 vs. 6.7	0.72 (95% CI, 0.62–0.83)	<0.001
**Second-line chemotherapy**									
Pelzer U, et al. (2011) [121]	CONKO-003	Locally advanced/metastatic	OFF versus BSC	23 vs. 23	NA	NA	4.82 vs. 2.30	0.45 (95% CI, 0.24–0.83)	0.008
Oettle H, et al. (2014) [122]	CONKO-003	Locally advanced/metastatic	OFF vversus 5-FU/LV	77 vs. 91	NA	3.0vs. 2.1	5.9 vs. 3.3	0.66 (95% CI, 0.48–0.91)	0.010
Gill S. et al. (2016) [123]	PANCREOX	Locally advanced/metastatic	mFOLFOX6 versus 5-FU/LV	54 vs. 54	13.2% vs. 8.5%	3.1 vs. 2.9	6.1 vs. 9.9	1.78 (95% CI, 1.08–2.93)	0.024
Wang-Gillam A, et al. (2016) [117]	NAPOLI-1	Metastatic	nal-IRI/5-FU/LV versus 5-FU/LV versus nal-IRI	117 vs. 149 vs. 151	16% vs. 1% vs. 6%	3.1 vs. 1.5 vs. 2.7	6.1 vs. 4.2 vs. 4.9	0.67 (95% CI, 0.49–0.92) 0.99 (95% CI, 0.77–1.28)	0.0120.94
**Precision medicine**									
Marabelle A, et al. (2020) [19]	KEYNOTE-158	Locally advanced/metastatic	Pembrolizumab	22	18%	2.1	4.0	NA	NA
Golan T, et al. (2019) [156]	POLO	Metastatic ^‡^	Olaparib versus placebo	92 vs. 62	23% vs. 12%	7.4 vs. 3.8	19.0 vs. 19.2 [131]	0.83 (95% CI, 0.56–1.22)	0.3487

ORR, objective response rate; PFS, progression-free survival; OS, overall survival; HR, hazard ratio; NA, not applicable; CI, confidence interval; FOLFIRINOX, 5-FU/LV/irinotecan/oxaliplatin; OFF, oxaliplatin/5-FU/leucovorin; BSC, best supportive care; LV, leucovorin; mFOLFOX6, 5-FU/LV/oxaliplatin; nal-IRI, nanoliposomal irinotecan; *, one-sided; ^†^, non-inferiority; ^‡^, maintenance therapy setting after first-line platinum-based chemotherapy.

## Data Availability

Not applicable.

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
