# Peer review of "Impact of Endoscopic Ultrasound-Guided Tissue Acquisition on Decision-Making in Precision Medicine for Pancreatic Cancer: Beyond Diagnosis"

_diagnostics, 2021, doi:10.3390/diagnostics11071195_

Round 1
Reviewer 1 Report
I read with great interest the review proposed by Imaoka et al, as it emphasizes the potential of EUS for for pancreatic cancer management. The authors focus on the precision medicine concept and its potential use for pancreatic cancer. While endoscopic ultrasound represents the standard for harvesting pancreatic tumor specimens, there are still challenges that might be discussed starting from what needle type should be used, diagnosis and genetic testing. The authors seem to follow a direction within this review begining from tissue acquisition by describing some steps, to biomarker, genetic testing, and an oncologic view with precision medicine.
The three main chapters of this review “EUS-guided tissue acquisition”, “biomarker testing”, “precision medicine for pancreatic cancer” provide either many basic details or specific information. I recommend shortening the last chapter as it focuses, and rephrasing some of the sentences so that the authors should specify better why specific genes should be tested and when.
A major drawback of this current study is that the authors did not discuss the potential of organoids for pancreatic cancer therapy, as it requires tissue acquisition and you may test certain drugs. I consider it worthy of appraisal and a sub-chapter should definitely be inserted.
As the authors are not native speakers, some of the ideas within the text are rather confusing and definitely needs rephrasing:
Line 40 “Furthermore, neither serologic evaluation nor abdominal imaging have disadvantage in difficulty of identifying tumors in an early stage” – this should be rephrased
Line 44 „The reported sensitivity and specificity of CA19-9 for the diagnosis of PC range from 70%-90% and 90%” – some citations are missing
Line 82 “in this review we mainly discuss” – the review should be easy to read, and objectives very well defined
Paragraph 174-182 should be rewritten as it is not really clear what the authors try to emphasize.
Line 226 “so also are reports” – should be rephrased
Line 251 “too long or too short a duration of fixation, inadequate concentration of the fixative, etc.) – should be rephrased
Line 257 guarantee replace with provide
Line 258 – 260 Typically, tissue samples acquired by EUS TA sometimes contain relatively fewer neoplastic cells as compared to surgical FFPE samples. Is it either typically or sometimes, the sentence is confusing
Line 266„In NGS performed in small specimens, the tumor fraction, the percentage of tumor cells in the tissue sample, is more important than celullarity”, please rephrase
Line 275 “may contribute to difficulties” please change it
Line 303 “It is generally not practical for testing” please find a more suitable idea
Line 309-311. The general idea is missing.
Line 401 proteins: CTLA-4, insert „are”
Line 415 the phrase is very confusing
In the precision medicine chapter, discussing some therapies which are not tested by EUS TA does not meet the objective of this review. Even more as the title of the manuscript states „impact of endoscopic ultrasound” the authors should definitely decide what they discuss throught or they may suggest possible future directions.
The „Conclusions” do not sustain the objectives and neither the entire the review and they should be expanded.
Please also include this study and discuss its potential.
Bruno, R.; Sensi, E.; Lupi, C.; Giordano, M.; Bernardini, L.; Vivaldi, C.; Fornaro, L.; Vasile, E.; Campani, D.; Fontanini, G. Feasibility of BRCA1/2 Testing of Formalin-Fixed and Paraffin-Embedded Pancreatic Tumor Samples: A Consecutive Clinical Series. Diagnostics 2021, 11, 1046. https://doi.org/10.3390/diagnostics11061046
Author Response
Dear Sir:
We wish to express our strong appreciation to the reviewers for their insightful comments on our paper. We feel the comments have helped us significantly improve this review. We will try to answer your concerns and issues. As a result, we made some corrections to the text in the manuscript, and added many references. We highlight the parts modified by red coloring in the manuscript.
Reviewer 1 Comment
Query-1
The three main chapters of this review “EUS-guided tissue acquisition”, “biomarker testing”, “precision medicine for pancreatic cancer” provide either many basic details or specific information. I recommend shortening the last chapter as it focuses, and rephrasing some of the sentences so that the authors should specify better why specific genes should be tested and when.
Response-1
We appreciate the reviewer`s advise. As reviewer suggestion, we extensively rewrote the last chapter, taking into account other advice (The modifications are extensive, so please see the last chapter indicated in red). However, some parts actually became longer as a result.
We also added references and added the following text in the section "3. Biomarker testing".
"The American Society of Clinical Oncology (ASCO) guidelines recommended early biomarker testing in PC patients who are likely to be potential candidates for precision medicine after first-line chemotherapy"
Now we believe that our manuscript has been improved.
Query-2
A major drawback of this current study is that the authors did not discuss the potential of organoids for pancreatic cancer therapy, as it requires tissue acquisition and you may test certain drugs. I consider it worthy of appraisal and a sub-chapter should definitely be inserted.
Response-2
We appreciate the reviewer`s suggestion. As reviewer suggestion, we added the following description about organoids in new sub-chapter "3.4. Future perspectives".
" Another future perspective is to apply EUS-TA to establishment of organoids. Organoids are three-dimensional self-assembled in vitro cultured structures derived from tissues and stem cells. They are widely used in various research fields, from basic developmental and stem cell research to personalized medicine. Recently, there have been increasing reports on the organoids establishment using tissue sample obtained by EUS-TA, and success rate of organoids establishment is ranged 70-82% 103 104 105 106. This patient-derived organoids have the potential to be used as drug screening and predicting the response to chemotherapeutic agents in pancreatic cancer patients in the future 107."
Now we believe that our manuscript has been more insightful.
Query-3
As the authors are not native speakers, some of the ideas within the text are rather confusing and definitely needs rephrasing:
- Line 40 “Furthermore, neither serologic evaluation nor abdominal imaging have disadvantage in difficulty of identifying tumors in an early stage” – this should be rephrased
→As reviewer suggestion, we changed the following text.
" Furthermore, neither serologic evaluation nor abdominal imaging have disadvantage in difficulty of identifying tumors in an early stage. "
to
" Furthermore, both serologic evaluation and abdominal imaging have disadvantage in difficulty of identifying tumors in an early stage. "
- Line 44 „The reported sensitivity and specificity of CA19-9 for the diagnosis of PC range from 70%-90% and 90%” – some citations are missing
→As reviewer suggestion, we added references (Ref 4-9).
- Line 82 “in this review we mainly discuss” – the review should be easy to read, and objectives very well defined
→As reviewer suggestion, we changed the following text in the section "1. Introduction".
" In this review, we mainly discuss tissue sampling by EUS-TA for biomarker testing, and the current status of precision medicine for PC."
to
" In this review, we summarize the recent advances in the application of EUS-TA to biomarker testing, and the current status of precision medicine for PC based on these biomarkers."
- Paragraph 174-182 should be rewritten as it is not really clear what the authors try to emphasize.
→As reviewer pointed out, we changed the following text in the section "2. Endoscopic ultrasound-guided tissue acquisition (EUS-TA)".
" According to these findings about EUS-TA, it is important to remain attentive to the need for obtaining tissue samples that would allow biomarker testing of the obtained specimens for the diagnosis of metastasis or recurrence several years after surgery. In such cases, use of a 22-gauge FNB needle and the non-suction technique may be optimal for tissue sampling for biomarker testing. On the other hand, in EUS-TA for preoperative cases or cases of recurrence after surgery, the primary aim of EUS-TA is the diagnosis of malignancy, and biomarker testing is not necessary in tissue samples obtained by EUS-TA; surgical specimens should be used for biomarker testing, as they provide more adequate samples."
to
" When performing EUS-TA, it is important to remain attentive to the need for obtaining tissue samples that would allow ancillary biomarker testing. In cases of metastasis, use of a 22-gauge FNB needle and the non-suction technique may be optimal for tissue sampling. On the other hand, in EUS-TA for preoperative cases or cases of recurrence after surgery, the primary aim of EUS-TA is the diagnosis of malignancy, and biomarker testing is not necessary in tissue samples obtained by EUS-TA; surgical specimens should be used for biomarker testing, as they provide more adequate samples."
- Line 226 “so also are reports” – should be rephrased
→As reviewer pointed out, we changed the following text in the section "3.3. Next-generation sequencing".
" Both the development of precision medicine and of NGS are dramatically changing the treatment of PC. Consequently, as the impact of EUS-TA is growing, so also are reports of NGS performed in tissue samples obtained by EUS-TA 68 69 70 71 48 72."
to
" NGS enables the sequencing of multiple genes in a limited number of samples obtained by EUS-TA. Recently, several studies have reported the adequacy of EUS-TA samples for NGS 61 86 87 84 64 88."
- Line 251 “too long or too short a duration of fixation, inadequate concentration of the fixative, etc.) – should be rephrased
→As reviewer pointed out, we changed the following text in the section "3.3. Next-generation sequencing".
" i.e., too long or too short a duration of fixation, inadequate concentration of the fixative, etc."
to
" i.e. inappropriate duration of fixation or concentration of fixative "
- Line 257 guarantee replace with provide
→As reviewer pointed out, we changed "guarantee" to "provide" in the section "3.3. Next-generation sequencing".
- Line 258 – 260 Typically, tissue samples acquired by EUS TA sometimes contain relatively fewer neoplastic cells as compared to surgical FFPE samples. Is it either typically or sometimes, the sentence is confusing
→As reviewer pointed out, we changed the following sentence in the section "3.3. Next-generation sequencing".
" Typically, tissue samples acquired by EUS-TA sometimes contain relatively fewer neoplastic cells as compared to surgical FFPE samples."
to
" Typically, tissue samples acquired by EUS-TA contain relatively fewer neoplastic cells as compared to surgical FFPE samples."
- Line 266„In NGS performed in small specimens, the tumor fraction, the percentage of tumor cells in the tissue sample, is more important than celullarity”, please rephrase
→As reviewer pointed out, we changed the following sentence in the section "3.3. Next-generation sequencing".
" In NGS performed in small specimens, the tumor fraction, the percentage of tumor cells in the tissue sample, is more important than cellularity."
to
" The tumor fraction, the percentage of tumor cells in the tissue sample, is more important than cellularity, when NGS is performed in small specimens. "
- Line 275 “may contribute to difficulties” please change it
→As reviewer pointed out, we changed the following sentence in the section "3.3. Next-generation sequencing".
" These components may contribute to difficulties in obtaining accurate results."
to
" These components may interfere with obtaining accurate results."
- Line 303 “It is generally not practical for testing” please find a more suitable idea
→As reviewer pointed out, we changed the following sentence in the section "3.3. Next-generation sequencing".
" WGS typically requires 100-1,000 ng of DNA. It is generally not practical for testing limited amounts of tissue samples obtained by EUS-TA."
to
" WGS typically requires 100-1,000 ng of DNA, and it is generally not practical for testing limited amounts of tissue samples obtained by EUS-TA. "
- Line 309-311. The general idea is missing.
→As reviewer pointed out, we changed the following sentence in the section "3.3.2 Whole-exome sequencing".
" allows researchers to focus on the part of the genome that is more likely to contain clinically actionable alterations, such as mutations of tumor-suppressor genes."
to
" Exome sequencing allows researchers to focus on the part of the genome that is more likely to contain clinically actionable alterations, such as mutations of tumor-suppressor genes. "
- Line 401 proteins: CTLA-4, insert „are”
→As reviewer pointed out, we changed the following sentence in the section "4.1. Immune checkpoint inhibitors".
" Currently, approved ICIs target immune checkpoint proteins: CTLA-4, PD-1, and PD-L1."
to
" Currently, approved ICIs target immune checkpoint proteins are CTLA-4, PD-1, and PD-L1. "
- Line 415 the phrase is very confusing
→As reviewer pointed out, we changed the following sentence in the section "4.1. Immune checkpoint inhibitors".
" For MSI-H/dMMR testing, MSI analysis, IHC for MMR proteins (MLH1, MSH2, MSH6, PMS2), and NGS are used. "
to
" MSI analysis, IHC for MMR proteins, and NGS are used for MSI-H/dMMR testing. "
Query-4
In the precision medicine chapter, discussing some therapies which are not tested by EUS TA does not meet the objective of this review. Even more as the title of the manuscript states „impact of endoscopic ultrasound” the authors should definitely decide what they discuss throught or they may suggest possible future directions.
Response-4
Thank you for reviewer's pointed out. As reviewer's pointed out, the indication of PARP inhibitors is not directly tested by EUS-TA now. On the other hand, EUS-TA findings may become important in the future, as shown in the paper reviewer's proposed in query-6. Therefore, we revised the following sentences in the section "4.2. PARP inhibitors".
" On the other hand, the effect of PARP inhibitors against PCs harboring somatic BRCA1/2 mutations are still unclear. Recently, one phase II study showed the efficacy of olaparib administered as monotherapy in previously treated PC patients with DNA damage repair gene (DDR) alterations other than germline BRCA1/2 mutations133. The overall efficacy was modest (ORR, 2%; median PFS, 3.7 months; OS, 13.6 months). However, olaparib was demonstrated to exert efficacy in 24 patients with DDR phenotypes (14 with ATM, 3 with ARID1A, 2 with PALB2, 6 with other alterations) (ORR, 18.2%; median PFS, 5.7 months; OS, 13.6 months). This study included 1 patient with a somatic BRCA mutation, but the antitumor effect in this patient was not described. Therefore, when BRCA1/2 mutations are found by NGS of tumor tissue specimens, it is necessary to distinguish between somatic and germline mutations. Germline alterations are not only detected in cancer tissues, but also in normal non-neoplastic tissues. Identification of germline mutations is important not only for predicting the response to olaparib and platinum-based chemotherapy, but also to predict the risk of familial cancer syndromes. "
to
" On the other hand, the effect of PARP inhibitors against PCs harboring somatic BRCA1/2 mutations are still unclear. However, one phase II study showed the efficacy of olaparib administered as monotherapy in previously treated PC patients with DNA damage repair gene (DDR) alterations other than germline BRCA1/2 mutations 157. The overall efficacy was modest (ORR, 2%; median PFS, 3.7 months; OS, 13.6 months). However, olaparib was demonstrated to exert efficacy in 24 patients with DDR phenotypes (14 with ATM, 3 with ARID1A, 2 with PALB2, 6 with other alterations). Another PARP inhibitor, rucaparib, has shown some activities in PC patients with a germline or somatic BRCA1/2 mutations in phase 2 trials. Shroff et al. reported the efficacy of rucaparib in previously treated PC patients with a germline or somatic BRCA1/2 mutations 158. A total 19 patients were enrolled in this trial (16 with germline mutation, 3 with somatic mutation). Although subsequent enrollment was stopped due to futility, 3 out of the 4 responders (2 showed CR and 2 showed PR). Rucaparib was also demonstrated to show efficacy as maintenance therapy after first-line platinum-based chemotherapy in PC patients with germline or somatic BRCA1/2, or germline PALB2 mutations159. Rucaparib showed response in 3 out of 6 patients with a germline PALB2 mutation, and 1 out of 2 patients with a somatic BRCA1/2 mutation. One meta-analysis reported that similar ORR of PARP inhibitor in patients with germline and somatic BRCA1/2 mutations 160. These findings indicate that PARP inhibitors may be effective in patients with a somatic BRCA1/2 mutation and other DDR alterations as well as in patients with a germline BRCA1/2mutation. In the near future, it may be meaningful to evaluate a somatic BRCA1/2 mutation as a biomarker. In such scenario, NGS using tissue samples would allow us to detect both germline and somatic BRCA1/2 mutations. When BRCA1/2 mutations are found by NGS of tumor tissue specimens, it is necessary to distinguish between somatic and germline mutations 161. "
Now we believe that our manuscript has been more insightful.
Query-5
The „Conclusions” do not sustain the objectives and neither the entire the review and they should be expanded.
Response-5
Thank you for reviewer's suggestion. As reviewer's suggestion, we changed the following sentences in the section "5. Conclusion".
"Precision medicine for PC remains challenging, and still benefits only a small population of patients. However, relatively promising results have been obtained in patients who EUS-TA enables identification of biomarkers even in the limited amount of tissue specimens available, thereby opening the path to precision medicine in patients with PC. Thus, while performing EUS-TA, the operator should aware of the possible use of the specimens submitted for biomarker testing."
to
" Precision medicine still benefits only a small population of patients with PC. However, biomarker-based therapy has shown promising results in patients who once had no treatment options. With the advancement of precision medicine for PC, the impact of EUS-TA on therapeutic decision-making will become more significant. Thus, while performing EUS-TA, the operator should remain attentive to the need for obtaining tissue samples that would allow biomarker testing."
Now we believe that our manuscript has been greatly improved.
Query-6
Please also include this study and discuss its potential.
Bruno, R.; Sensi, E.; Lupi, C.; Giordano, M.; Bernardini, L.; Vivaldi, C.; Fornaro, L.; Vasile, E.; Campani, D.; Fontanini, G. Feasibility of BRCA1/2 Testing of Formalin-Fixed and Paraffin-Embedded Pancreatic Tumor Samples: A Consecutive Clinical Series. Diagnostics 2021, 11, 1046. https://doi.org/10.3390/diagnostics11061046
Response-6
We appreciate the reviewer`s suggestion. This study is very important for our review. We not only cited it, but also extensively rewrote the "4.2. PARP inhibitors" section extensively to reflect the content of this study.
" Germline mutations of BRCA1/2 are known to be associated with an increased risk of development of ovarian cancer and breast cancer 121 122 123, and are also found in PCs at a frequency of about 4%-7% 124 125 126 127 128. Presence of germline BRCA1/2 mutations is reported to be associated with enhanced antitumor effects of platinum drugs, such as oxaliplatin, which exert their antitumor effects by inducing the formation of DNA interstrand cross-links and DNA damage 129. PARP inhibitors, such as olaparib, inhibit homologous recombination during the repair process after DNA damage, and thus exert their antitumor effects by inducing apoptosis of tumor cells harboring germline BRCA1/2 mutations. Olaparib has already been shown to be useful for maintenance treatment after platinum-based chemotherapy of ovarian cancer harboring germline/somatic BRCA1/2 mutations 130 10.
In a phase III study (POLO trial), olaparib was demonstrated to show efficacy as maintenance therapy after first-line platinum-based chemotherapy in PC patients with germline BRCA1/2 mutations 131. The PFS, the primary endpoint of the trial, was 7.4 months in the olaparib group and 3.8 months in the placebo group (hazard ratio [HR], 0.53; 95% confidence interval [95% CI], 0.35-0.82). Among the major grade 3 or higher toxicities, anemia, fatigue or asthenia, and decreased appetite were observed in 11%, 5%, and 3% of patients in the olaparib group, and 3%, 2%, and 0% of patients in the placebo group, respectively. A final analysis of the OS showed no difference between the olaparib group and the placebo group (19.0 months vs. 19.2 months; HR, 0.83; 95% CI, 0.56-1.22) 132. However, this result is not surprising, considering that a larger trial population is required to obtain a significant difference in the OS and that patients from the placebo group were allowed to cross over after the trial.
On the other hand, the effect of PARP inhibitors against PCs harboring somatic BRCA1/2 mutations are still unclear. Recently, one phase II study showed the efficacy of olaparib administered as monotherapy in previously treated PC patients with DNA damage repair gene (DDR) alterations other than germline BRCA1/2 mutations133. The overall efficacy was modest (ORR, 2%; median PFS, 3.7 months; OS, 13.6 months). However, olaparib was demonstrated to exert efficacy in 24 patients with DDR phenotypes (14 with ATM, 3 with ARID1A, 2 with PALB2, 6 with other alterations) (ORR, 18.2%; median PFS, 5.7 months; OS, 13.6 months). This study included 1 patient with a somatic BRCA mutation, but the antitumor effect in this patient was not described. Therefore, when BRCA1/2 mutations are found by NGS of tumor tissue specimens, it is necessary to distinguish between somatic and germline mutations. Germline alterations are not only detected in cancer tissues, but also in normal non-neoplastic tissues. Identification of germline mutations is important not only for predicting the response to olaparib and platinum-based chemotherapy, but also to predict the risk of familial cancer syndromes.
For identification of germline alterations, peripheral blood specimens are the most frequently used biological material. In addition, the variant allele frequency (VAF) reported by NGS can aid in distinguishing germline mutations from somatic mutations. A mutation is potentially a germline mutation if the VAF is approximately 50% (heterozygous) or 100% (homozygous). The VAF values associated with somatic mutations are usually lower, because these mutations are not present in all cells. However, somatic mutations may be associated with VAF values of over 50%, if the tumor fraction in the analyzed sample is high 134 135."
to
" Germline mutations of BRCA1/2 are known to be associated with an increased risk of development of ovarian cancer and breast cancer 146 147 148, and are also found in PCs at a frequency of about 4%-7% 149 150 151 152 153. Presence of germline BRCA1/2 mutations is reported to be associated with enhanced antitumor effects of platinum drugs, such as oxaliplatin, which exert their antitumor effects by inducing the formation of DNA interstrand cross-links and DNA damage 154. PARP inhibitors, such as olaparib, inhibit homologous recombination during the repair process after DNA damage, and thus exert their antitumor effects by inducing apoptosis of tumor cells harboring germline BRCA1/2 mutations. Olaparib has already been shown to be useful for maintenance treatment after platinum-based chemotherapy of ovarian cancer harboring germline/somatic BRCA1/2 mutations 155 16.
In a phase III study (POLO trial), olaparib was demonstrated to show efficacy as maintenance therapy after first-line platinum-based chemotherapy in PC patients with germline BRCA1/2 mutations 156. The PFS, the primary endpoint of the trial, was 7.4 months in the olaparib group and 3.8 months in the placebo group (hazard ratio [HR], 0.53; 95% confidence interval [95% CI], 0.35-0.82).
On the other hand, the effect of PARP inhibitors against PCs harboring somatic BRCA1/2 mutations are still unclear. However, one phase II study showed the efficacy of olaparib administered as monotherapy in previously treated PC patients with DNA damage repair gene (DDR) alterations other than germline BRCA1/2 mutations 157. The overall efficacy was modest (ORR, 2%; median PFS, 3.7 months; OS, 13.6 months). However, olaparib was demonstrated to exert efficacy in 24 patients with DDR phenotypes (14 with ATM, 3 with ARID1A, 2 with PALB2, 6 with other alterations). Another PARP inhibitor, rucaparib, has shown some activities in PC patients with a germline or somatic BRCA1/2 mutations in phase 2 trials. Shroff et al. reported the efficacy of rucaparib in previously treated PC patients with a germline or somatic BRCA1/2 mutations 158. A total 19 patients were enrolled in this trial (16 with germline mutation, 3 with somatic mutation). Although subsequent enrollment was stopped due to futility, 3 out of the 4 responders (2 showed CR and 2 showed PR). Rucaparib was also demonstrated to show efficacy as maintenance therapy after first-line platinum-based chemotherapy in PC patients with germline or somatic BRCA1/2, or germline PALB2 mutations159. Rucaparib showed response in 3 out of 6 patients with a germline PALB2 mutation, and 1 out of 2 patients with a somatic BRCA1/2 mutation. One meta-analysis reported that similar ORR of PARP inhibitor in patients with germline and somatic BRCA1/2 mutations 160. These findings indicate that PARP inhibitors may be effective in patients with a somatic BRCA1/2 mutation and other DDR alterations as well as in patients with a germline BRCA1/2mutation. In the near future, it may be meaningful to evaluate a somatic BRCA1/2 mutation as a biomarker. In such scenario, NGS using tissue samples would allow us to detect both germline and somatic BRCA1/2 mutations. When BRCA1/2 mutations are found by NGS of tumor tissue specimens, it is necessary to distinguish between somatic and germline mutations 161. For identification of germline alterations, peripheral blood specimens are the most frequently used biological material. However, the variant allele frequency (VAF) reported by NGS can aid in distinguishing germline mutations from somatic mutations. A mutation is potentially a germline mutation if the VAF is approximately 50% (heterozygous) or 100% (homozygous). The VAF values associated with somatic mutations are usually lower, because these mutations are not present in all cells. However, somatic mutations may be associated with VAF values of over 50%, if the tumor fraction in the analyzed sample is high 162 163."
Now we believe that our manuscript has been greatly improved. Thank you.

Reviewer 2 Report
This is a clinical review on a very important and hot topic: precision medicine for pancreatic cancer (PDAC). Because approximately 80% of patients with PDAC will never undergo resection, tissue collected with endoscopic ultrasound-tissue acquisition is the only one available for molecular diagnostics.
The manuscript is well written but, from an endoscopic point of view, I suggest reducing/summarizing the chapter about the suction vs. non-suction techniques, which is not an important point. Differently, you should implement the chapter on FNA vs FNB because these two techniques could have different outcomes on NGS and molecular diagnostics. In the following points I’ll raise some important concepts that you could include and discuss, adding an amount of important literature:
- Line 125: the sentence “the superiority of EUS-FNB needles remains under debate” should be changed. Indeed, the sentence is true for the comparisons between FNA needles and side-fenestrated needles (i.e., ProCore). However, new generation end-cutting needles demonstrated to outperform FNA ones (PMID: 32162287 + your ref. 45).
- Moreover, end-cutting needles (SharkCore and Acquire) outperformed also the side-fenestrated ones (PMID: 32433914, PMID: 32408361, and PMID: 32583392).
- This is true not only for PDAC but also for pNET (PMID: 33390343 and PMID: 31579710). This point should be mentioned because PDAC diagnosis is not made “a priori”.
- Therefore, EUS-TA is moving from FNA to FNB (PMID: 29941722).
- This clinical practice change impacts also the feasibility of NGS as demonstrating in this randomized trial (PMID: 31367675) and these retrospective studies (PMID: 31070037; PMID: 30422342).
- Finally, NGS on FNB specimens demonstrated to be reliable compared with matched surgical specimens (PMID: 32205062).
- However, another point you should discuss, is that cytological samples from EUS‐FNA are rich in pure tumor cells compared with histological specimens, which are often rich in tumor stroma. This could impair the feasibility of NGS in some FNB specimens. Furthermore, FNA smear alcohol‐based fixation improves the preservation of nucleic acids that are partially degraded by formalin‐fixation of histologic specimens (PMID: 29024530). This drawback can be overcome using the touch imprint cytology, which has been demonstrated to provide high-quality cytological samples from FNB specimens, thus obtaining both valuable cytological and histological samples using a single FNB needle (PMID: 30484917)
Minor:
Please spell EUS-FNA at its first appearance in the text (line 89).
Author Response
Dear Sir:
We wish to express our strong appreciation to the reviewers for their insightful comments on our paper. We feel the comments have helped us significantly improve this review. We will try to answer your concerns and issues. As a result, we made some corrections to the text in the manuscript, and added many references. We highlight the parts modified by red coloring in the manuscript.
Reviewer 2 Comment
Major
Query-1
The manuscript is well written but, from an endoscopic point of view, I suggest reducing/summarizing the chapter about the suction vs. non-suction techniques, which is not an important point. Differently, you should implement the chapter on FNA vs FNB because these two techniques could have different outcomes on NGS and molecular diagnostics. In the following points I’ll raise some important concepts that you could include and discuss, adding an amount of important literature:
Reponse-1
We appreciate the reviewer`s suggestion. As reviewer's said, the chapter about the suction vs. non-suction techniques seems somewhat redundant. Thus, we reduced and changed the following text in the section "2.3. Technical aspects (Suction vs. Non-suction technique)".
" In the conventional EUS-FNA procedure, after needle puncture of the target lesions, the typical technique includes stylet removal followed by the use of to-and-fro motions using negative-pressure suction. It was initially expected that the negative-pressure suction would increase the amount of tissue specimen obtained, but in fact, it only increased the amount of blood contamination 49 50. In contrast to the conventional suction technique using a syringe, the slow-pull technique, which was recently introduced for EUS-TA, minimizes the negative pressure, because the stylet itself is removed from the needle slowly and continuously. Theoretically, this slow-pull technique can minimize blood contamination and allow higher quality specimens to be obtained. However, the improved diagnostic performance of the non-suction technique still remains under debate. As compared to the conventional suction technique, there are reports that the non-suction technique is associated with a significantly lower rate of blood contamination 51 52 and higher cellularity 53. On the other hand, many reports suggest the absence of any significant difference in the diagnostic performance between the two techniques 51 54 53 55 52 56. One meta-analysis concluded that, although there was no difference in the accuracy or sensitivity between the two techniques, use of the non-suction technique was associated with a significantly lower rate of blood contamination as compared to use of the conventional suction technique 57. "
to
" In contrast to the conventional suction technique using a syringe, the slow-pull technique, which was recently introduced for EUS-TA, minimizes the negative pressure, because the stylet itself is removed from the needle slowly and continuously. It was expected that slow-pull technique would minimize blood contamination and allow higher quality specimens to be obtained. However, the improved diagnostic performance of this non-suction technique still remains under debate. As compared to the conventional suction technique, there are reports that the non-suction technique is associated with a significantly lower rate of blood contamination 65 66 and higher cellularity 67. On the other hand, many reports suggest the absence of any significant difference in the diagnostic performance between the two techniques 65 68 67 69 66 70. One meta-analysis concluded that, although there was no difference in the accuracy or sensitivity between the two techniques, use of the non-suction technique was associated with a significantly lower rate of blood contamination as compared to use of the conventional suction technique 71."
Furthermore, in order to implement the chapter on FNA vs FNB, we changed the following text in the section "2.1. Needle type (EUS-FNA vs. EUS-FNB)".
(1) Line 125: the sentence “the superiority of EUS-FNB needles remains under debate” should be changed. Indeed, the sentence is true for the comparisons between FNA needles and side-fenestrated needles (i.e., ProCore). However, new generation end-cutting needles demonstrated to outperform FNA ones (PMID: 32162287 + your ref. 45).
→As reviewer pointed out, we added references and changed the following text in the section "2.1. Needle type (EUS-FNA vs. EUS-FNB)".
"Despite several clinical trials and meta-analyses conducted to compare the diagnostic performance between EUS-FNA and -FNB needles, the superiority of EUS-FNB needles still remains under debate 35 36 37 38 39 40 41 42 43 44. On the other hand, in a recent randomized controlled trial of a 22-gauge FNB needle versus 22-gauge FNA needle, the FNB needle showed a higher histologic core procurement rate than the FNA needle 45."
to
" Although the diagnostic benefit of EUS-FNB needles had been debated for a long time 41 45 46 47 48 49 50 51 52 53 44, the diagnostic accuracy of third-generation EUS-FNB needles has shown superiority in accuracy in recent randomized controlled trials 54 55.Furthermore, the FNB needle showed a higher histologic core procurement rate than the FNA needle 54. "
(2) Moreover, end-cutting needles (SharkCore and Acquire) outperformed also the side-fenestrated ones (PMID: 32433914, PMID: 32408361, and PMID: 32583392).
→As reviewer pointed out, we added references and the following text in the section "2.1. Needle type (EUS-FNA vs. EUS-FNB)" to make this point clearer.
" Therefore, EUS-FNB needles began to be developed to preserve the tissue architecture and improve the sample adequacy and diagnostic accuracy. The first developed EUS-FNB needle was stiff and difficult to use. However, newer generation EUS-FNB needles was developed and improved the sample adequacy and diagnostic accuracy 42 43 44."
(3) This is true not only for PDAC but also for pNET (PMID: 33390343 and PMID: 31579710). This point should be mentioned because PDAC diagnosis is not made “a priori”.
→As reviewer suggestion, we added references and changed the following text in the section "2.1. Needle type (EUS-FNA vs. EUS-FNB)".
" This is also true for pancreatic neuroendocrine tumor 56 57."
(4) Therefore, EUS-TA is moving from FNA to FNB (PMID: 29941722).
→As reviewer suggestion, we added references and changed the following text in the section "2.1. Needle type (EUS-FNA vs. EUS-FNB)".
" To date, the superiority of the EUS-FNB needle in tissue sample quality has been widely accepted 58. "
(5) This clinical practice change impacts also the feasibility of NGS as demonstrating in this randomized trial (PMID: 31367675) and these retrospective studies (PMID: 31070037; PMID: 30422342).
→As reviewer suggestion, we added references and added the following text in the section "2.1. Needle type (EUS-FNA vs. EUS-FNB)".
" Furthermore, considering tissue sample quality, EUS-FNB needle may be more suitable for biomarker testing as shown in the randomized trial 59 and retrospective studies 60 61."
(6) Finally, NGS on FNB specimens demonstrated to be reliable compared with matched surgical specimens (PMID: 32205062).
→As reviewer suggestion, we added references and added the following text in the section "3. Biomarker testing".
"Several studies have examined the surrogacy of EUS-TA samples for surgically resected specimens in NGS. The results of these studies of NGS using EUS-TA samples showed high concordance with that in a surgically resected specimen used as a reference standard 82 83 84. These findings are important because PC patients who potentially who benefit from precision medicine are ineligible for surgical resection"
(7) However, another point you should discuss, is that cytological samples from EUS‐FNA are rich in pure tumor cells compared with histological specimens, which are often rich in tumor stroma. This could impair the feasibility of NGS in some FNB specimens. Furthermore, FNA smear alcohol‐based fixation improves the preservation of nucleic acids that are partially degraded by formalin‐fixation of histologic specimens (PMID: 29024530). This drawback can be overcome using the touch imprint cytology, which has been demonstrated to provide high-quality cytological samples from FNB specimens, thus obtaining both valuable cytological and histological samples using a single FNB needle (PMID: 30484917)
→ We appreciate the reviewer`s valuable advise. As reviewer suggestion, we added references and the following sentences in the section "3.4. Future perspectives".
" Future perspectives in NGS is the use of cytological specimens. Cytological smear specimens of EUS-FNA are rich in tumor cells, whereas FFPE specimens are often rich in tumor stroma. Recently, the excellent performance of cytological smear specimens in molecular analyses has been shown 98 99. In lung cancer, the College of American Pathologists guidelines state that pathologists may use either FFPE samples (cell blocks) or other cytological preparations (cytological smears) as suitable specimens for biomarker molecular testing, including NGS 100. In addition, FNA rinse sample and touch imprint cytology are promising approaches for NGS. EUS-TA samples are routinely rinsed and fixed in an alcohol-based fixative immediately after their acquisition, and this contributes to excellent preservation and quality of DNA and RNA. Wei et al. compared the performance of this FNA rinse samples and that of cell block samples used for NGS 101. NGS was successfully performed using all samples, but much more DNA was obtained from the FNA rinse samples compared with the paired cell block samples (176.3 vs. 10.6 ng/μL, respectively). Touch imprint cytology is simple and cost-effective method: tissue sample is touch onto a slide, leaving an imprint on the glass slide in the form of cells. Crinó et al. reported that sample quality of touch imprint cytology was comparable to cytological smear specimens 102. Touch imprint cytology allows obtaining both valuable cytological smear and histological samples in a single session of EUS-FNB. Thus, this method is a potentially useful method for NGS."
Now we believe that our manuscript has been greatly improved.
Minor
Query-1
Please spell EUS-FNA at its first appearance in the text (line 89).
Response-1
Thank you for pointing this out. I've added explanations of the abbreviations (EUS-FNA and EUS-FNB).

Round 2
Reviewer 1 Report
Authors made the necessary changes, thus, I recommend it for publication.
Reviewer 2 Report
Thank you for revising the manuscript. It is now suitable for publication. I have just one more minor request for the Authors. Recently, a very important randomized trial has been published and deserve to be included in this review (PMID: 34116031).
Good luck with publication!